# Comprehensive Research on Past and Future Therapeutic Strategies Devoted to Treatment of Amyotrophic Lateral Sclerosis

**DOI:** 10.3390/ijms23052400

**Published:** 2022-02-22

**Authors:** Belgin Sever, Halilibrahim Ciftci, Hasan DeMirci, Hilal Sever, Firdevs Ocak, Burak Yulug, Hiroshi Tateishi, Takahisa Tateishi, Masami Otsuka, Mikako Fujita, Ayşe Nazlı Başak

**Affiliations:** 1Department of Pharmaceutical Chemistry, Faculty of Pharmacy, Anadolu University, Eskisehir 26470, Turkey; belginsever@anadolu.edu.tr; 2Medicinal and Biological Chemistry Science Farm Joint Research Laboratory, Faculty of Life Sciences, Kumamoto University, Kumamoto 862-0973, Japan; hiciftci@kumamoto-u.ac.jp (H.C.); htateishi@kumamoto-u.ac.jp (H.T.); motsuka@gpo.kumamoto-u.ac.jp (M.O.); 3Department of Drug Discovery, Science Farm Ltd., Kumamoto 862-0976, Japan; 4Department of Molecular Biology and Genetics, Koc University, Istanbul 34450, Turkey; hdemirci@ku.edu.tr; 5Ministry of Health, Istanbul Training and Research Hospital, Physical Medicine and Rehabilitation Clinic, Istanbul 34098, Turkey; severhil@gmail.com; 6Faculty of Medicine, Kocaeli University, Kocaeli 41001, Turkey; 200601195@kocaeli.edu.tr; 7Department of Neurology and Neuroscience, Faculty of Medicine, Alaaddin Keykubat University, Alanya 07425, Turkey; burak.yulug@alanya.edu.tr; 8Division of Respirology, Neurology and Rheumatology, Department of Medicine, Kurume University School of Medicine, Fukuoka 830-0011, Japan; tateishi_takahisa@med.kurume-u.ac.jp; 9Suna and İnan Kıraç Foundation, Neurodegeneration Research Laboratory (KUTTAM-NDAL), Koc University, Istanbul 34450, Turkey

**Keywords:** amyotrophic lateral sclerosis (ALS), oxidative stress, protein aggregation, glutamate excitotoxicity, apoptosis, neuroinflammation, axonal degeneration, edaravone, riluzole, induced pluripotent stem cells (iPSCs)

## Abstract

Amyotrophic lateral sclerosis (ALS) is a rapidly debilitating fatal neurodegenerative disorder, causing muscle atrophy and weakness, which leads to paralysis and eventual death. ALS has a multifaceted nature affected by many pathological mechanisms, including oxidative stress (also via protein aggregation), mitochondrial dysfunction, glutamate-induced excitotoxicity, apoptosis, neuroinflammation, axonal degeneration, skeletal muscle deterioration and viruses. This complexity is a major obstacle in defeating ALS. At present, riluzole and edaravone are the only drugs that have passed clinical trials for the treatment of ALS, notwithstanding that they showed modest benefits in a limited population of ALS. A dextromethorphan hydrobromide and quinidine sulfate combination was also approved to treat pseudobulbar affect (PBA) in the course of ALS. Globally, there is a struggle to prevent or alleviate the symptoms of this neurodegenerative disease, including implementation of antisense oligonucleotides (ASOs), induced pluripotent stem cells (iPSCs), CRISPR-9/Cas technique, non-invasive brain stimulation (NIBS) or ALS-on-a-chip technology. Additionally, researchers have synthesized and screened new compounds to be effective in ALS beyond the drug repurposing strategy. Despite all these efforts, ALS treatment is largely limited to palliative care, and there is a strong need for new therapeutics to be developed. This review focuses on and discusses which therapeutic strategies have been followed so far and what can be done in the future for the treatment of ALS.

## 1. Introduction

Amyotrophic Lateral Sclerosis (ALS), also known as Lou Gehrig’s or Charcot disease, is characterized by progressive deterioration of the upper and lower motor neurons in the brain and spinal cord, which leads to muscle weakness, paralysis and, finally, death due to respiratory failure within three to five years after the onset of the symptoms. ALS generally starts in limb or bulbar muscles, then spreads to other body parts and culminates in respiratory muscle dysfunction [1,2]. The primary symptoms of ALS due to motor neuronal degeneration are fasciculation, muscle cramps and stiffness, dysarthria, dysphagia, emotional lability (pseudobulbar affect (PBA)), which is characterized by uncontrolled laughter or crying, and dyspnea. Furthermore, sialorrhea usually occurs in ALS patients owing to dysphagia, and increased saliva production can result in aspiration pneumonia. In general, ALS patients also suffer from depression and anxiety [3,4].

An orphan or rare disease is defined as a condition that affects a small percentage of the population [5]. The incidence of ALS was reported to be between 0.6 and 3.8 per 100,000 persons-years [6,7]. Therefore, the United States Food and Drug Administration (FDA) consequently recognised ALS as an orphan disease [8]. Approximately 90% of ALS cases are sporadic (sALS), and the remaining 10% cases are familial (fALS) [9]. In their clinical presentation, fALS and sALS are indistinguishable. Many environmental factors have been investigated in sALS. Advanced age (between 55 and 65 years) is an established risk factor, though fALS is generally related to a younger age of onset. Male gender, smoking, physical stress, exposure to heavy metals, persistent pollutants and environmental toxins such as β-methylamino-L-alanine, which is why people in Guam were susceptible to ALS with higher prevalence [10,11], are other risk factors that trigger sALS [12,13,14]. However, the genetic background, the etiology and the exact mechanisms of sALS are not yet known. The identification of genes responsible for fALS is shedding further light on the disease mechanisms underlying sALS [15]. In fALS, there are more than 25 utterly different genes participating in the familial disease, and this number is increasing. Point mutations in superoxide dismutase 1 (SOD1, 20% of all fALS) and expanded GGGGCC repeats in C9orf72 are the most frequent ALS genes, whereas mutations in TAR deoxyribonucleic acid (DNA) binding protein 43 (TARDBP), fused in sarcoma/translocated in liposarcoma (FUS/TLS), TANK-binding kinase 1 (TBK1) or valosin-containing protein (VCP) account for less common genetic causes [16,17]. Interestingly, the existence of abnormal phosphorylated and ubiquitinated aggregates of TDP-43 is detected in approximately 97% of sALS patients, except for SOD1 and FUS cases [18]. 

The first animal model of ALS was a SOD1^G93A^ transgenic mouse model, generated based on a SOD1 mouse, harboring the glycine to alanine change at amino acid residue 93 (G93A) [19,20]. This transgenic model overexpressing mutant SOD1 is the most commonly used animal model in ALS for the screening of drug compounds at preclinical level. However, the majority of these compounds failed in human clinical trials, despite the promising outcomes obtained from preclinical studies [21,22]. Alternatively, mammalian cell lines transfected with the SOD1 mutants have been used in numerous studies (NSC-34 mouse neural hybrid cell line and SH-SY5Y human neuroblastoma cell line, e.g., [23,24,25]). 

Oxidative stress, mitochondrial dysfunction, glutamate-induced excitotoxicity, apoptosis, neuroinflammation, axonal degeneration, skeletal muscle deterioration and viruses are other proven crucial pathologic mechanisms (Figure 1) associated with the formation of ALS, indicating that ALS is multifactorial syndrome rather than a single disease [26,27].

There is also a report that the gut microbiota, composed of bacteria, viruses, fungi and other microorganisms, and CNS disease-modifying bioactive metabolites derived from gastrointestinal tract can contribute to the pathogenesis of ALS [29]. In addition, a significant decrease in the expression of tight junction protein zona occludens protein 1 (ZO-1) and the adherens junction protein E-cadherin were observed in the intestine of SOD1^G93A^ALS mice [30]. Other studies conducted with ALS patients also supported that gut microbiota could be a risk factor for ALS [31,32,33]. 

The blood–brain barrier (BBB), blood-spinal cord barrier (BSCB) and blood–cerebrospinal fluid barrier (BCSFB) play profound roles in the transfer of substances between the blood and brain/spinal cord. Any pathologic alteration in the BBB directly limits the passage of drugs, antibodies, gene carriers and stem cells from the capillaries to the CNS. Disintegration of the BBB and BSCB in areas of motor neuron injury in SOD1^G93A^ mice was observed at both early and late stages of ALS [34]. Abrahao et al. [35] deployed a first human trial relevant to the BBB opening in ALS patients using magnetic resonance-guided focused ultrasound (MRgFUS), and their results emphasized that this opening system could be coupled with the systemic administration of the most promising ALS therapeutics to directly target the degenerating motor cortex.

To date, edaravone (a free radical scavenger) and riluzole (a glutamate antagonist) are the only two drugs approved for use in the treatment of ALS, in spite of their limited beneficial effects on disease progression. The majority of other drugs have failed at phase III of clinical trials, which were performed with a larger cohort of ALS patients. Dextromethorphan hydrobromide and quinidine sulfate (Nuedexta^®^) also received approval from the FDA to treat PBA. These outcomes indicate that the discovery of efficient disease-modifying therapies is an urgent need for the treatment of ALS [36,37].

The discovery of induced pluripotent stem cells (iPSCs), analogous to embryonic stem cells (ESCs) with their pluripotency and in vitro self-renewal capability, is one of the major breakthroughs holding great promise for the treatment of both fALS and sALS. iPSCs can differentiate into specific cell phenotypes, such as motor neurons and astrocytes, the main targets in ALS pathophysiology. Motor neurons generated from iPSCs derived from somatic cells (usually peripheral blood mononuclear cells (PBMCs) or skin fibroblasts) of ALS patients have provided a highly innovative technique for modelling the disease. Besides, compared to ESCs, iPSCs are generated from patients’ own somatic cells, and thus there is a lack of ethical issues, decreased risk of immune rejection and limited immunosuppressive drug consumption. It is also beneficial that iPSCs retain the donor’s genetic information, whereas animal models do not fully reflect the unique features of the human nervous system. On the other hand, the potential risk over time for iPSCs derived from patients is the probability of occurrence of a similar degeneration after transplantation. The progress in CRISPR-9/Cas technique, a genome editing tool, has facilitated the improvement in gene mutations in iPSCs, derived from patients, and the appropriate comparison between the model and individuals [38,39,40,41,42]. Wang et al. [43] generated iPSCs from fibroblasts of fALS patients harbouring mutations and gene-corrected ALS iPSCs using the CRISPR/Cas9 system. iPSC-based techniques provide great insight for drug discovery research on ALS. Based on iPSC-based technologies, bosutinib, ropinirole and retigabine have been identified as promising anti-ALS drugs, for which clinical trials are ongoing [15]. Furthermore, an ALS-on-a-chip technology using the 3D-model of the motor unit derived from iPSCs was developed to provide a platform for screening drug candidates and researching the pathogenesis of ALS [44].

Another approach for ALS treatment is using non-invasive brain stimulation (NIBS) techniques, which regulate brain activity by means of different energy forms, such as electrical current, magnetic pulses or focused ultrasound given throughout the scalp and skull. Some studies about the employment of NIBS to ALS were reviewed, and these techniques could prove beneficial for ALS after the standardization of stimulation procedures [45]. 

Based on aforementioned data, the clinical and pathogenic heterogeneity of ALS and the absence of well-characterized and controlled cellular or animal models that recapitulate the disease dreadfully restrained the success of the drug development process. Therefore, it is obvious that a single therapeutic approach is inadequate for people living with ALS. In this review, we focus on some drugs and related candidates which were expected to target diverse mechanisms responsible for ALS pathogenesis and examine preclinical and clinical trials. Additionally, some combinatorial approaches such as molecular hybridization, drug repurposing and iPSCs-based phenotypic screening are discussed in order to be useful for future applications in ALS drug discovery.

## 2. Therapeutic Strategies for ALS Targets

### 2.1. Therapeutic Strategies against Oxidative Stress

Oxidative stress is the consequence of the imbalance between the generation of reactive oxygen species (ROS), such as hydrogen peroxide (H_2_O_2_), superoxide anions (O_2_^−^) and hydroxyl radicals (OH^−^), and the ability of antioxidant defence system to clean or repair the existing damage to proteins and/or DNA. Normally, SOD1 converts the superoxide anion to H_2_O_2_, but as the SOD1 mutation shows lower affinity for Zn^2+^, it donates an electron to O_2_ to generate O_2_^−^ at the Cu^2+^ catalytic site as a stronger oxidant and reacts with nitric oxide (NO) to form peroxynitrite (ONOO^−^), which is very detrimental to CNS. ROS is considered to be a major mechanism in ALS causing motor neuron death, whereas some studies imply that ROS only exacerbates disease progression. When glutamate receptors are over-activated in the presence of higher glutamate levels in the synapse, elevated calcium influx into the cell also triggers the entry of calcium into the mitochondria, causing mitochondrial dysfunction, further ROS production and, ultimately, cell death. Glial and infiltrated immune cells also abundantly contribute to the production of ROS because glial synapses, which surround the devastated neurons, stimulate glutamate excitotoxicity and elevated calcium entry into the cell and mitochondria. As a result, each process in neuronal degeneration repeats itself, like a vicious circle [46,47,48,49].

Since lipids constitute polyunsaturated fatty acids, they are prone to oxidize to peroxyl radicals, resulting in cerebral ischemia. As phenol derivatives are well-known radical scavenging agents, Mitsubishi Tanabe Pharma Corporation developed potential phenol-like radical scavengers for the treatment of cerebral infarction. They designed compounds carrying a carbonyl group, which can easily convert to hydroxy group by keto-enol tautomerization to achieve radical scavenging activity, similar to that of the phenol provided by its hydroxy group (Figure 2a). They identified edaravone as an effective radical scavenger among a variety of compounds. Edaravone is capable of scavenging both lipid- and water-soluble peroxyl radicals as well as ONOO^−^ among many types of ROS. Researchers also reported that the in vitro lipid peroxidation inhibitory activity of edaravone was reduced with the substitution of polar or hydrophilic groups in the 2-pyrazoline-5-one ring, while this activity was increased with the lipophilic substitutions on the phenyl ring (Figure 2b) [50,51]. Edaravone interacts with both peroxyl (LOO^ꞏ^) and hydroxyl (^ꞏ^OH) radicals by means of its enolate form (**B**), followed by the formation of a stable oxidation product (OPB: 2-oxo-3-(phenylhydrazono)-butanoic acid) through a radical intermediate (Figure 3) [52].

Edaravone (MCI-186, 3-methyl-1-phenyl-2-pyrazoline-5-one, Radicut^®^, Radicava^®^), a neuroprotective antioxidant drug, was synthesized via the reaction of phenylhydrazine and acetoacetic ester by Knorr in 1883 [51]. This procedure was also applied in other studies, as follows: Phenylhydrazine and methyl acetoacetate were mixed in the presence of glacial acetic acid and stirred at reflux for 3 h. The reaction mixture was evaporated to dryness and the residue was extracted with a mixture of water and ethyl acetate. The organic layer was separated, while the aqueous layer was further extracted with ethyl acetate. The combined organic layer was dried over anhydrous sodium sulphate and concentrated over reduced pressure to afford a pure product as a white solid (Figure 4) [53,54]. Edaravone (Figure 5) has been used to treat acute-phase cerebral infarction for almost 20 years in Japan. It received its approval for the treatment of ALS in Japan and South Korea in 2015; the FDA approved the drug in 2017 and Chinese–NMPA in 2019. In a phase II trial, 30 mg or 60 mg of edaravone was administered to 20 subjects with ALS. It was observed that the decline in the amyotrophic lateral sclerosis functional rating scale (ALSFRS-R) score was significantly reduced during the six-month treatment period [55]. In a randomised, double-blind phase III study, a significantly smaller decline in the ALSFRS-R score compared with placebo group was also observed [56]. The exact mechanism of action of edaravone for ALS remains unclear; it has both neuroprotective effects against oxidative stress and anti-inflammatory properties against activated microglial cells. So far, there has been no oral dosage formulation of edaravone clinically, thus it is used as an intravenous therapeutic for ALS management. However, oral dosage of edaravone for ALS patients is performed by Mitsubishi Tanabe Pharma Corporation under a phase 3b, multicentre, randomized, double-blind test, which has enrolled 380 ALS patients to examine and compare the efficacy of two dosing regimens of oral edaravone (Clinicaltrial.gov NCT04569084) [57,58,59]. 

Dopamine receptor agonists used for the treatment of Parkinson’s disease (PD) were also considered to be effective in ALS treatment. Pramipexole (Figure 5) has been shown to be a neuroprotective, reducing oxidative stress [60]. Dexpramipexole (Figure 5) is the optical enantiomer of pramipexole, with less affinity to dopamine receptors. Both compounds were screened in SOD1^G93A^ mice models, and dexpramipexole extended survival time and protected motor functions significantly more as compared to pramipexole [61]. In a randomised, double-blind, phase III trial in ALS, dexpramipexole was generally well-tolerated, but no beneficial outcome was determined [62] in spite of a significant benefit in a phase II study with ALS patients [63]. Bromocriptine (Figure 5) is a free-radical scavenger and a dopamine receptor agonist used in the treatment of PD. It was reported that bromocriptine protected motor neurons from oxidative injury in SOD1^H46R^ ALS mice models [64]. Furthermore, the results of a phase IIa, randomized, double-blind, placebo-controlled study conducted with Japanese ALS patients indicated that bromocriptine sustained motoneuronal activity, at least in part, by bromocriptine treatment, and this study stands out as promising for future phase IIb or III clinical trials [65]. Another dopamine receptor agonist used in the treatment of PD, ropinirole (Figure 5), was identified as a potential therapeutic candidate for ALS treatment from a large number of agents screened for the evaluation of multiple-phenotype rescue of sub-classified sALS models generated by using iPSCs [66]. A randomized, double-blind, placebo-controlled, single-center and open-label phase I/IIa clinical trial (UMIN000034954) is ongoing for ropinirole hydrochloride to test safety, tolerance and efficacy in ALS [67].

A monoamine oxidase-B (MAO-B) inhibitor used to treat PD, rasagiline (Figure 5), has been proposed to protect motor neurons in ALS patients thanks to its antioxidative and anti-apoptotic properties [68]. Rasagiline displayed a dose-dependent therapeutic effect on both motor activity and survival in SOD1^G93A^ models of fALS, alone or in cotreatment with riluzole [69]. No difference in the primary outcome of survival was observed between groups in a randomised, double-blind, parallel-group, placebo-controlled, phase II trial in ALS conducted with rasagiline [70]. Besides, rasagiline was observed not to alter the disease progression compared with controls over a year of treatment in a randomized, double-blind, placebo-controlled trial enrolling 80 ALS participants [71]. Selegiline (Figure 5), another MAO-B inhibitor with antioxidant properties, was also evaluated in a double-blind, placebo-controlled trial for treatment of ALS, but no significant effect was observed on the clinical progression of ALS [72].

Melatonin (Figure 5) is a potent antioxidant, anti-apoptotic and neuro-protectant agent [73]. In a combined study including cultured NSC-34 cells, SOD1^G93A^ mouse models and sALS clinical models enrolling 31 ALS patients demonstrated that in melatonin-reduced glutamate-induced cell death in NSC-34 cells, high-dose oral melatonin slowed disease progression and prolonged survival of SOD1^G93A^ mice. Furthermore, chronic high-dose (300 mg/day) rectal melatonin was well-tolerated in ALS patients [74]. In another recent study, a decline in annualized hazard death and a slower rate of decline in ALSFRS-R score was detected in melatonin users when compared with the non-melatonin users [75].

### 2.2. Therapeutic Strategies against Oxidative Stress via Protein Aggregation

SOD1 is a Cu/Zn-dependent antioxidant enzyme that catalyses the conversion of superoxide radicals to oxygen and hydrogen peroxide, thus regulating the superoxide levels that arise from mitochondrial inter-membrane space, cytosol and peroxisome. The discovery of the connection between mutations in the SOD1 gene with certain forms of fALS made a tremendous impact in understanding the pathology of ALS. SOD1 mutations have been reported to contribute to ALS through not only protein misfolding and aggregation, but also proteasome impairment, oxidative stress, oligodendrocyte degeneration and mitochondrial dysfunction [76,77]. 

In order to decrease SOD1 expression and increase SOD1 consumption, antisense oligonucleotides (ASOs) and heat shock protein (HSP) inducers were pursued, respectively. ASOs are short, synthetic nucleic acids which are suitable for chemical modification for high stability in biological fluids and potency in binding their mRNA target. ASOs targeting SOD1 mRNA reduced both SOD1 protein and mRNA levels throughout the brain and spinal cord in the SOD1^G93A^ mice model [78]. Tofersen (BIIB067, ISIS 333611, IONIS-SOD1Rx) is a type of antisense therapy. A randomised, placebo-controlled, first-in-man phase I clinical trial with SOD1-positive ALS patients indicated that ISIS 333611 was well-tolerated when administered as an intrathecal infusion [79]. A second phase I/II trial with BIIB067 was designed to decrease SOD1 mRNA in ALS patients carrying a SOD1 gene mutation. Results showed that a statistically significant reduction in cerebrospinal fluid (CSF)–SOD1 suggested substantial reduction in CNS tissue SOD1. Besides reduced CSF phosphorylated heavy neurofilament, a decline in ALSFRS-R scores were also detected in ALS patients [80]. A phase III trial of BIIB067 is currently ongoing (Clinicaltrial.gov NCT02623699).

HSPs such as Hsp27, Hsp40 and Hsp70 were associated with mutant SOD1 aggregates in SOD1 rodent models of ALS. Arimoclomol (Figure 6), a hydroxylamine-based HSP amplifier, was determined to extend survival and delay disease onset with improvement in neuromuscular activity in the SOD1^G93A^ mouse model of ALS [81,82]. A double-blind, randomized, placebo-controlled phase II trial enrolling patients with rapidly progressive SOD1-mutant ALS suggested a possible therapeutic benefit of arimoclomol [83]. A randomized, double blind, placebo-controlled phase III trial has been completed, but results are not yet posted (Clinicaltrial.gov NCT03491462) [84]. Furthermore, HSPB8, a specific chaperone, was found to be effective in stimulating the clearance of the misfolded proteins related to motor neuron diseases, such as mutant SOD1 and a TDP-43 fragment in ALS. Colchicine (Figure 6), an anti-inflammatory drug generally used in gout treatment, enhances the expression of HSPB8. Colchicine was shown to reduce the aggregation of TDP-43 misfolded species responsible for motor neuron death in sALS [85]. A randomised, double-blind, placebo-controlled, multicentre phase II clinical trial is ongoing for colchicine (Clinicaltrial.gov NCT03693781) [86].

Cu^II^(atsm) (Figure 6) is a copper carrying compound which is capable of crossing the BBB, and it has been developed as a potential new anti-ALS agent. Cu^II^(atsm) treatment aims to maintain bioavailable copper to mutant SOD1 to keep it in a stable physiological holo form, as mutant SOD1 aggregates are observed at abnormal copper-deficient state [87]. The preclinical potential of Cu^II^(atsm) in ALS was investigated in the SOD1^G37R^ mouse model. According to the results, Cu^II^(atsm) enhanced locomotor performance and prolonged mouse survival in a dose-dependent manner [88]. Phase II/III testing is in progress after successful completion of the phase I trial in ALS patients. In this assay involving post-mortem spinal cord tissue from sALS, it is considered that Cu^II^(atsm) can be beneficial to sALS, enabling copper to transfer from the complex into mutant SOD1 [89]. 

The antimalarial drug pyrimethamine (Figure 6) was identified by high-throughput screening (HTS) to lower SOD1 protein levels in human cells in a dose-dependent manner. Therefore, a phase I study enrolling 16 ALS patients harbouring SOD1 mutations was performed. These findings supported the connection of a reduction in SOD1 in ALS patients with pyrimethamine use. However, the significance is not yet fully understood [90]. Another study also demonstrated that pyrimethamine caused a significant reduction in total CSF mutant SOD1 levels in ALS patients in addition to its safety and well-tolerated profile, though the molecular target by which pyrimethamine decreases the SOD1 amount is not yet known [91]. 

Capper et al. [92] reported that the cysteine-reactive molecule ebselen (Figure 6) was able to engage with Cys111, which is prone to oxidative modifications in SOD1-ALS. Ebselen forms a covalent bond with SOD1-Cys111 and repairs the monomer–dimer equilibrium of SOD1^A4V^ to wildtype, acting as a potent bifunctional pharmacological chaperone for SOD1 endowed with combinatory properties of human copper chaperone for SOD1 (hCCS) and edaravone. 

### 2.3. Therapeutic Strategies against Mitochondrial Dysfunction

Mitochondria are essential cell organelles that elicit adenosine triphosphate (ATP) through oxidative phosphorylation, regulate calcium homeostasis and produce ROS as a by-product of the electron transport chain. Dysfunction of this electron transport chain can cause increased levels of mitochondrial oxygen consumption and ROS production along with decreased ATP synthesis and DNA repair. Mitochondrial dysfunction also triggers oxidative stress and glutamate excitotoxicity in ALS. Abnormal morphology and ALS-linked mutant proteins were observed in mitochondria of sALS post-mortem tissue [47,48,93].

Olesoxime (TRO19622) (Figure 7) is a mitochondrial pore modulator with a cholesterol-like structure. Bordet et al. [94] determined that olesoxime in vitro promoted motor neuron survival among a collection of a great number of low molecular-weight compounds which were capable of preventing motor neuron cell death. This compound was also found to improve motor performance, delay the onset of the clinical disease and extend survival in SOD1^G93A^ transgenic mice. They also suggested that olesoxime displayed neuroprotective activity, binding to the mitochondrial permeability transition pore (mPTP). However, a phase II–III trial of olesoxime in subjects with ALS revealed that olesoxime did not show a significant beneficial effect in ALS patients treated with riluzole [95].

Creatine (Figure 7), phosphorylated by creatine kinase, is involved in ATP production. Therefore, creatine stabilizes the mitochondrial creatine kinase and inhibits opening of the mPTP. It was detected that creatine improved motor performance and extended survival in SOD1^G93A^ mice [96]. However, a double-blind, placebo-controlled, sequential clinical trial established that creatine monohydrate showed no beneficial effect on survival or disease progression in ALS patients [97]. Creatine also displayed no benefit in any outcome measure in a randomized double-blind, placebo-controlled trial on 104 ALS patients [98] or in another multicentre, double-blinded study with 107 ALS patients [99]. Coenzyme Q10 (CoQ10) (Figure 7), an electron acceptor in the mitochondrial respiratory chain, was reported to increase lifespan in SOD1^G93A^ mice [100]. In a two-stage, bias-adjusted, randomized, placebo-controlled, double-blind, phase II design performed for CoQ10 in 185 ALS patients, a decline in ALSFRS-R score and insufficient data were observed for further phase III testing [101].

### 2.4. Therapeutic Strategies against Glutamate-Induced Excitotoxicity

It has long been known that abnormal levels of glutamate, which is the main excitatory neurotransmitter in the brain, can cause neurodegenerative effects. The α-amino-3-hidroxy-5-methyl-4-isoxazole-propionic acid (AMPA) and the *N*-methyl-D-aspartate (NMDA) receptors, two important ionotropic glutamate receptors, are mediators of glutamate excitotoxicity owing to poor buffering calcium influx capacity of their subunits, such as the GluA2 subunit of the AMPA receptor. Massive entry of calcium into the cell stimulates phospholipases, proteases and endonucleases, causing devastation of energy metabolism and apoptotic or necrotic cell death. Besides, reduced expression of astrocytic excitatory amino-acid transporters (EAATs) is crucial for clearance of glutamate from the synaptic cleft into astrocytes, and defects in glutamate transport have been linked with ALS pathogenesis [102,103,104].

Riluzole (6-(trifluoromethoxy)-2-aminobenzothiazole) was discovered due to its remarkable anticonvulsant properties during the 1980s. The reaction of 4-(trifluoromethoxy)aniline with tetrabutylammonium thiocyanate afforded riluzole, in the presence of dichloromethane, with a yield of 61%. An alternative synthetic route for riluzole is the reaction of 4-(trifluoromethoxy)aniline with ammonium thiocyanate in the presence of acetonitrile (71%) (Figure 8) [105]. Riluzole (Figure 9) has been reported to exhibit neuroprotective effects in several animal models of PD, Huntington’s disease and cerebral ischemia [106]. Moreover, two large randomized placebo-controlled clinical trials were carried out to investigate the potency of riluzole in ALS treatment [107]. A double-blind, placebo-controlled phase II trial enrolling 155 ALS outpatients showed that riluzole slowed the disease progression and improve survival in patients [108]. In order to confirm and extend the therapeutic effect of riluzole, a double-blind, placebo-controlled, multicentre, international, dose-ranging study was performed on a large number of ALS outpatients. Results showed that it was well-tolerated and diminished the risk of death or tracheostomy in ALS patients [109]. Riluzole was approved by the FDA in 1995 and subsequently launched into the market under the trade name, Rilutek^®^. Riluzole, the first FDA-approved drug for the treatment of ALS, showed its mechanism via enhancement of extracellular glutamate uptake and inhibition of glutamate release from presynaptic terminals. Furthermore, riluzole also stabilizes the inactivated state of voltage-dependent sodium channels and NMDA receptor-mediated responses [110]. On the other hand, riluzole, dexpramipexole and pramipexole are 1,3-benzothiazole-based derivatives, though they display their anti-ALS activities in a different manner, implying the importance of the compounds carrying the same structural moiety endowed with different functions [111].

Gabapentin (Figure 9), a lipophilic analogue of γ-aminobutyric acid (GABA), is an antiepileptic drug and a modulator of the glutamatergic system. Due to its diminished effects on glutamate excitotoxicity, it was considered to be effective in ALS patients. After successful results of preclinical and clinical studies of gabapentin, a phase III randomized clinical trial was also undertaken. However, there were no data obtained showing any beneficial effect on symptoms and disease progression in ALS patients [112,113].

As an AMPA receptor antagonist and an antiepileptic drug, topiramate (Figure 9), was determined to maintain motor neurons from glutamate excitotoxicity in a dose-dependent manner in an in vitro model. However, a controlled clinical trial exhibited that the maximum tolerated dosage of topiramate was related to severe adverse effects and absence of efficacy [114,115]. Another AMPA receptor antagonist, perampanel (Figure 9), shows its effect via the inhibition of the excitation of postsynaptic membranes. It was deduced that it slowed the development of the ALS phenotype and boosted the number of anterior horn cells in transgenic mice [116]. However, an open label pilot study, which was conducted to search for the tolerability and safety of perampanel in ALS patients, was terminated owing to the high number of adverse events [117]. Perampanel was also determined to cause a decline in ALSFRS-R score, along with several adverse events, in a randomized, double-blind, placebo-controlled, multicentre, phase II clinical study [118].

On the other hand, talampanel (Figure 9) is a non-competitive AMPA receptor blocker and belongs to a structural class of substituted benzodiazepines. It was observed from a study in a mutant SOD1 mouse model of ALS that talampanel could be beneficial in the earlier stages of the disease [119]. Moreover, in talampanel-treated subjects, declines in muscle strength and in the ALSFRS-R score were found to be 15% and 30% less, respectively, in a double-blind, placebo-controlled, multicenter, randomized phase II trial. This study also indicated that talampanel was sufficiently well-tolerated in subjects with ALS for further a phase III trial [120]. 

Memantine (Figure 9) is a non-competitive NMDA receptor antagonist marketed for advanced stages of Alzheimer’s disease. It was revealed that memantine delayed the disease progression and increased the life span of SOD1^G93A^ mice [121]. In a phase II/III, 12-month, double-blinded, single-centre, randomized clinical trial, which was carried out to determine the efficacy and safety of memantine in ALS, the tolerance and safety profile of memantine in ALS patients was confirmed, however, no evidence of efficacy was obtained [122]. A new multicentred double blind and placebo-controlled study is ongoing (Clinicaltrial.gov NCT02118727) to determine whether combination therapy of memantine with riluzole can improve the progression of the disease, along with cognitive deficits. Dextromethorphan (Figure 9) is an NMDA-glutamate receptor antagonist that also displays cough suppressant properties. A randomized, double-blind, placebo-controlled study enrolling 45 ALS patients substantiated that a relatively low dose of dextromethorphan revealed no recovery in one year survival in ALS [123]. Moreover, quinidine (Figure 9) can enable high plasma dextromethorphan concentrations, preventing its metabolism. Therefore, the combination of dextromethorphan hydrobromide and quinidine sulfate (Figure 9) was found to regulate symptoms of PBA acting on the Sigma-1 receptor and is approved by the FDA for the treatment of PBA in patients with ALS [124,125].

Intriguingly, ceftriaxone (Figure 9), a member of third generation β-lactam antibiotics, was reported to intercept glutamate-induced excitotoxicity by elevating the expression and functional activity of EAAT2. In a study conducted with SOD1^G93A^ mice, ceftriaxone postponed the loss of neurons and extended survival time [126]. Promising results were achieved at stages I and II of a three-stage randomised, double-blind, placebo-controlled study of ceftriaxone in ALS; however, ceftriaxone displayed no clinical efficacy at stage III [127]. In a structural manner, ceftriaxone shares a similar lactam ring and an exocyclic amide group with bromocriptine, a free-radical scavenger, despite their distinct mechanisms (reducing glutamate excitotoxicity and oxidative stress, respectively), suggesting that the lactam ring or exocyclic amide group could be important regions for anti-ALS function [128]. 

Calcium channel blockers were also considered to be potent in ALS treatment by reducing the downstream effects of glutamate excitotoxicity by means of remittance of calcium entry into damaged neurons. A phase II clinical study revealed that verapamil (Figure 9) was ineffective in retarding the progression of ALS [129]. Another randomized, placebo-controlled, double-blind crossover study enrolling 87 ALS patients revealed that nimodipine (Figure 9) was also found ineffective in slowing the progression of ALS [130]. On the other hand, mexiletine (Figure 9) has potential for lowering hyperexcitability as a sodium channel blocker. Therefore, it underwent clinical trials for ALS treatment. A single blind, randomized, controlled phase II trial of mexiletine concluded that a daily 300 mg dose of mexiletine had no effects on axonal sodium current and ALSFRS-R score [131]. A different randomized trial of mexiletine in ALS indicated that mexiletine also had no effect on rate of progression when administered daily to ALS patients as 300 mg and 900 mg doses [132]. Another randomized, double-blind, crossover trial indicated that 150 mg mexiletine twice daily was well-tolerated and effective for controlling the symptom of muscle cramps in ALS [133]. A phenyltriazine-based antiepileptic agent, lamotrigine (Figure 9), inhibits both sodium and calcium channels and hampers glutamate presynaptic release. According to a double-blind, placebo-controlled, crossover study, no clinical effect of lamotrigine on ALS progression was detected [134,135,136]. Retigabine (ezogabine) (Figure 9) is an anticonvulsant drug which activates a voltage-gated potassium-channel. Retigabine was found to reduce hyperexcitability and extend motor neuron survival in iPSC-derived motor neurons from ALS patients harbouring SOD1 mutations [137]. A phase II randomized clinical trial of retigabine enrolling 65 ALS patients showed that retigabine diminished cortical and spinal motor neuron excitability in a dose-dependent manner [138].

The vitamin B_12_ analog, methylcobalamin (Figure 9), was reported to possess neuroprotective properties towards glutamate-induced cytotoxicity [139]. It was anticipated, in a study conducted in the wobbler mouse model of ALS, that methylcobalamin would show improvements in motor symptoms at very high dose levels [140]. Although a long-term phase II/III randomised controlled study enrolling 373 ALS patients revealed less decline in the ALSFRS-R score along with low treatment-related adverse events, there was no significant efficacy in the whole cohort. This treatment could extend survival and decrease symptomatic progression without major side effects if started early [141].

### 2.5. Therapeutic Strategies for Reducing Apoptosis and/or Boosting Autophagy

Apoptosis is another potential mechanism of motor neuron death in ALS. It is termed as suicidal death of cells with a certain morphology, whereas autophagy is lysozyme-mediated sequestration of cytoplasmic material in vacuoles for bulk degradation. Autophagy is responsible for removing protein aggregates, damaged or redundant organelles through mitochondria or the endoplasmic reticulum along with toxic metabolites, cancer cells and intact microorganisms. Due to their lack of proliferation capacity and their high energy consumption, autophagy is essential for neurons to remove protein aggregates and damaged cellular inclusions. The Bcl-2 family proteins, as suppressors (Bcl-2, Bcl-XL) or promoters (Bax, Bad, Bak and Bcl-xS), regulate apoptosis, and it has recently become clear that they also control autophagy [142,143].

The Abelson non-receptor tyrosine kinase (c-Abl, Abl1) is normally activated in the Bcr–Abl hybrid protein of chronic myelogenous leukemia (CML). In addition to its oncogenic potential, overexpression of active c-Abl induces apoptosis and cell cycle arrest in response to a wide range of stimuli, such as inflammation, DNA damage, amyloid-β and oxidative stress, resulting in neurodegeneration and neuroinflammation [144,145]. It was also rationally hypothesized that activation of the Src/c-Abl signalling pathway synergistically leads to apoptosis in non-dividing cells (motor neurons) and the secretion of inflammatory factors in dividing cells (astrocytes and microglia) (Figure 10) [146]. There is growing evidence that the c-Abl pathway is a therapeutic target in ALS, based on the studies in cell culture revealing that inhibition of c-Abl protects cortical neurons from DNA damage-induced apoptosis [147]. Moreover, a robust rise in the phosphorylation of c-Abl in the brain and spinal cord of symptomatic SOD1^G93A^ mice and in the spinal cord and motor cortex of symptomatic SOD1^G86R^ mice was observed [148]. It was also documented that c-Abl expression was increased three-fold in post-mortem spinal cord tissues from sALS patients compared with non-ALS patients [149].

Katsumata et al. [149] designated that mutations of SOD1 activated the upregulation of c-Abl and alleviated cell viability. Additionally, dasatinib (Figure 11), a BBB-permeable c-Abl inhibitor, prevented cytotoxicity of mutant SOD1 proteins. Activation of c-Abl and caspase-3 was observed in SOD1^G93A^ mice. Dasatinib extended the survival of SOD1^G93A^ mice and diminished c-Abl phosphorylation. Imatinib (Figure 11), the c-Abl inhibitor used to treat CML, was reported to inhibit c-Abl phosphorylation in primary neuronal cultures at micromolar levels in response to various stimuli, such as oxidative stress. Imatinib was also determined to prevent astrocyte conditioned media (ACM)-hSOD1^G93A^-mediated motoneuron death [148]. Imamura et al. [150] determined that Src/c-Abl could be an efficient target for ALS treatment, based on HTS conducted for 1416 compounds, using ALS survival of motor neurons generated from an ALS patient harbouring SOD1 mutations. Bosutinib (Figure 11), a Src/c-Abl inhibitor used for treatment of CML, increased autophagy and reduced the levels of misfolded SOD1. Similar results were obtained with other mutations including TDP-43 and C9ORF72. Furthermore, bosutinib treatment modestly increased the life span of a mutant SOD1 ALS mice model. Bosutinib is currently in an open-label, multicentre phase I trial for ALS (UMIN000036295) to evaluate its safety and tolerability for the treatment of ALS patients and to explore its efficacy on ALS [151]. 

A hydrophilic bile acid used for treatment of chronic cholestatic liver disorders, tauroursodeoxycholic acid (TUDCA) (Figure 11), was identified to inhibit apoptosis in ventral mesencephalic tissue cultures and within the transplants [152]. A double-blind placebo-controlled trial enrolling 34 ALS patients determined that TUDCA was well-tolerated and, at study end, the baseline-adjusted ALSFRS-R score was found to be higher in TUDCA-treated cohorts than in placebo cohorts [153]. A phase III clinical trial for TUDCA with more participants is ongoing (Clinicaltrial.gov NCT03800524). 

Rapamycin (Sirolimus^®^) (Figure 11) is used clinically to hamper solid organ transplant rejection. It is also identified as a mammalian target of rapamycin (mTOR) inhibitor and consequent autophagy activator [154]. In regard to these properties, rapamycin discernibly counteracted accumulation of misfolded proteins and exhibited beneficial effects in studies using different cell lines and ALS animal models. A randomized, double-blind, placebo controlled, multicenter phase II clinical trial continues for rapamycin with ALS patients (Clinicaltrial.gov NCT03359538) [155]. Lithium is a glycogen synthase kinase-3 (GSK-3) inhibitor and is currently used to treat bipolar disorder. Moreover, lithium is an inositol monophosphatase 1 (IMPA 1) inhibitor, and thus a mTOR-independent autophagy inducer [154]. It was reported that lithium administration to SOD1^G93A^ mice promoted life span and activated autophagy [156]. A randomised, double-blind, placebo-controlled trial in ALS patients indicated no beneficial effect of lithium on survival and safety concerns [157]. Another double-blind, placebo-controlled trial with a combination of lithium with riluzole did not alleviate the progression of ALS more than riluzole alone [158]. Valproic acid (Figure 11), an antiepileptic drug, is a histone deacetylase (HDAC) and GSK-3 inhibitor. It also has antiapoptotic properties through upregulation of Bcl-2 protein. A study including the treatment SOD1^G93A^ mice with valproic acid accentuated a significant prolongation of the disease’s duration [159]. Conversely, a randomized sequential study of valproic acid in ALS patients provided no advantage to survival or disease progression [160]. As the cotreatment, lithium with valproic acid was found to halt glutamate excitotoxicity; this synergistic effect was considered to be effective in ALS [161]. A study performed with SOD1^G93A^ mice demonstrated that this combined treatment resulted in longer lifespan compared to monotreatment with lithium or valproic acid [162]. In a clinical experiment, cotreatment of valproic acid and lithium significantly boosted survival and neuroprotection in sALS patients beyond late adverse events [163].

### 2.6. Therapeutic Strategies against Neuroinflammation

Neuroinflammation is another important pathological mechanism in ALS progression. It is characterized by the activation of microglia and astrocytes, infiltration of immune cells (lymphocytes, macrophages, mast cells, neutrophils, etc.) and higher levels of inflammatory mediators. Microglia are generally divided into inflammatory (M1) and activated (M2) phenotypes. The M1 is linked with the release of proinflammatory cytokines (tumour necrosis factor (TNF)-α, interleukin-6 (IL-6), IL-23, IL-1β, IL-12, NO), cytotoxic substances (ROS), prostaglandin E2, chemokines, dysregulated glutamate levels, whereas M2 expresses anti-inflammatory molecules such as IL-10 and transforming growth factor (TGF)-β. There is mounting evidence regarding the connection between neuroinflammation with neuronal loss in sALS and fALS. Initially, glia and T cells, especially M2 macrophages/microglia, and T helper (Th) 2 cells and regulatory T (Treg) cells have protective roles in maintaining motor neuron viability. At a later stage, cytotoxic M1 macrophages/microglia and proinflammatory Th1 and Th17 T cells become more active [164,165,166,167,168].

Tocilizumab is an antibody that inhibits signalling of IL-6, a well-known promoter of the development of Th17 cells. It is currently used in patients with rheumatoid arthritis. A pilot study documented that in vitro tocilizumab mitigated several factors, including IL-6, that drive inflammation in sALS patients [169]. Another study including in vivo baseline inflammatory gene transcription in PBMCs of sALS patients indicated that tocilizumab infusions partially normalized inflammation of sALS patients [170]. A placebo-controlled phase II study with ALS patients who were genotyped for Asp^358^Ala polymorphism of the IL-6 receptor gene showed that tocilizumab treatment was safe and well-tolerated and alleviated C-reactive protein (CRP) levels in CSF relevant to IL-6 receptor Asp^358^Ala genotype. However, there was no difference in PBMC gene expression or clinical measures between groups [171]. 

Due to the efficacy of IL-1 inhibition in ALS, a single arm pilot study was performed for anakinra (Figure 12), an inhibitor of IL-1 receptor, in ALS patients. This study resulted in decreased levels of cytokines and the inflammatory marker fibrinogen during the first 24 weeks of treatment, but not in a significant reduction in disease progression [172].

A colony stimulating factor 1R receptor (CSF1R) and c-kit inhibitor, masitinib (Figure 12), is effective against microglial, macrophage and mast cell activation. It has been given the orphan drug status and screened for treatment of various disorders [173,174]. Masitinib restrained CSF-induced proliferation, cell migration and the expression of inflammatory mediators in microglia cultures from symptomatic SOD1^G93A^ spinal cords [175] and post paralysis treatment of SOD1^G93A^ mice. Masitinib also abolished mast cell and neutrophil infiltration and axonal pathology [176]. Masitinib was shown to be effective in ALS at 4.5 mg/kg/d according to a double-blind study with 394 randomly-assigned ALS patients. A confirmatory phase III study will be started to confirm these findings [177].

Ibudilast (MN-166) (Figure 12), an inhibitor of macrophage migration inhibitory factor and phosphodiesterases 3, 4, 10 and 11, is capable of suppressing neuronal cell death induced by microglial activation. It was found to increase the removal of TDP-43 and SOD1 protein aggregates in transfected cellular models of ALS [178]. ALS patients who underwent ibudilast treatment up to 100 mg/day for 36 weeks demonstrated no significant reduction in motor cortical glial activation analysed by PBR28-PET SUVR over 12–24 weeks and CNS neuroaxonal loss in an open label trial [179]. 

An approved immunomodulating drug for the treatment of multiple sclerosis (MS), fingolimod (Gilenya^®^) (Figure 12), shows its effects by blocking egress of lymphocytes from secondary lymph organs and reducing circulating lymphocytes associated with sphingosine 1-phosphate inhibition. A phase IIa trial of ALS verified that fingolimod was safe and well-tolerated and might reduce circulating lymphocytes in ALS patients. However, this trial had no statistical power to determine the effect of fingolimod on the ALSFRS-R score [180]. 

### 2.7. Therapeutic Strategies against Axonal Degeneration 

The membrane protein Nogo-A is an inhibitor of neurite outgrowth that was initially identified as a potent myelin-associated inhibitor of axonal growth and regeneration [181]. Overexpressed, high amounts of Nogo-A were detected in skeletal muscles of ALS-linked mutant SOD1 mice and in patients with sALS [182]. Nogo-A causes retrograde axonal degeneration by destabilizing the neuromuscular junction in ALS [183]. Ozanezumab, an anti-Nogo-A monoclonal antibody, was found well-tolerated at single and repeat dose administration in a randomized, first-in-human clinical trial [184]. However, ozanezumab was not found to be potent, compared with placebo, in ALS patients in a double-blind, placebo-controlled, phase II trial [185]. 

Rho kinase (ROCK) plays an important role in the formation of high levels of actin filaments and a reduced actin turnover, leading to destruction of cell growth and axonal regeneration. Therefore, ROCK inhibition is a promising approach for ALS treatment [186]. Fasudil (Figure 13), an important ROCK inhibitor, was ascertained to slow disease progression, improve survival time and attenuate motor neuron loss in SOD1^G93A^ mice [187]. A multicenter, double-blind, randomized, placebo-controlled phase IIa trial of fasudil in ALS patients also aims to assess the safety, tolerability and efficacy of fasudil at two different doses [188].

### 2.8. Therapeutic Strategies against Skeletal Muscle Deterioration

Degenerated motor neurons result in progressive muscle atrophy in ALS. Fast skeletal muscle troponin activators (FSTA) such as tirasemtiv and reldesemtiv stimulate the troponin complex, increasing its susceptibility to calcium and modulate muscle contraction [84,183,189].

Tirasemtiv (Figure 14) was found to enhance rotarod performance in B6SJL-SOD1^G93A^ transgenic mice with functional impairment [190]. In a randomized, double-blind, placebo-controlled trial on ALS patients, no treatment effect was ascertained for tirasemtiv, whereas slow vital capacity (SVC) and muscle strength declined at less than half the rate on tirasemtiv [191]. A phase III trial on ALS patients indicated that tirasemtiv showed no effect on the decline of SVC or secondary outcome measures. In spite of this unexpected outcome, the underlying mechanism of action was investigated with reldesemtiv (Figure 14), another fast skeletal muscle troponin activator [192]. A phase II, double-blind, randomized, dose-ranging trial on patients with ALS indicated that reldesemtiv was well-tolerated and stable against incremental rates of decline across multiple measures of ALS progression. For this reason, a phase III trial was planned for reldesemtiv [193].

### 2.9. Therapeutic Strategies against Viruses

Apart from the aforementioned factors, new categories such as infectious agents, including viruses, bacteria and fungi, have been recently considered. Among these agents, there is mounting evidence reporting the connection between viruses and ALS pathogenesis. Increased nonspecific reverse transcriptase activity in the blood and CSF of ALS patients compared to relatives and controls also supports this connection. In particular, retroviruses such as human immunodeficiency virus (HIV) and human T-cell leukemia virus type-1 can cause ALS-like syndrome. Hence, antiretroviral therapy could be a potential approach for retarding the symptoms of the ALS-like syndrome associated with HIV infection [194,195,196,197].

A combination of nucleoside reverse-transcriptase inhibitors lamivudine and abacavir and an HIV-1 integrase strand transfer inhibitor dolutegravir (Triumeq^®^) (Figure 15) was considered to be effective in ALS treatment. An open-label phase IIa trial conducted in ALS patients revealed that long-term exposure to this combination was safe and well-tolerated in this cohort. A larger international phase III trial will be performed to investigate the effect of this combination on overall survival and ALS progression [198]. 

### 2.10. The Importance of Trophic Factors in ALS

Trophic factors are essential for the enhancement of neuronal growth, survival and differentiation [199]. Growth factors are important for preventing motor neuron death in patients with ALS [200]. Due to its neurotrophic properties on motor neurons and the neuromuscular junction, recombinant human insulin-like growth factor I (rhIGF-I) was investigated for the determination of the relationship between rhIGF-I concentration and the prognosis of ALS. It was found that higher IGF-1 concentrations had the potential to increase survival [201]. Vascular endothelial growth factor (VEGF), a prominent factor for angiogenesis and neuroprotection, not only slowed the progression of ALS, but also increased life expectancy in SOD1^G93A^ mice [202]. Another substantial neurotrophic factor, hepatocyte growth factor (HGF), alleviated motoneuron death and axonal degeneration and extended the life span of SOD1^G93A^ mice [203]. A phase I study of HGF for ALS was performed from 2011 to 2015 at Tohoku University, Japan, and a phase II study commenced in May 2016 [204]. 

Intriguingly, epidermal growth factor receptor (EGFR) mRNA was detected to be overexpressed 10-fold more in the spinal cord of patients with ALS, just as in SOD1^G93A^ transgenic mice model. Therefore, erlotinib (Figure 16), an important EGFR inhibitor, in particular for the treatment of EGFR mutated advanced or metastatic non-small-cell lung carcinoma (NSCLC), was tested on SOD1^G93A^ mice. However, results exhibited that erlotinib could not extend survival, though it delayed the onset of multiple behavioural measures of disease progression [205].

### 2.11. Newly Synthesized and Evaluated Compounds as Anti-ALS Agents

Chen et al. [206] defined two arylsulfanyl pyrazolone (ASP) bearing compounds, 5-((2,4-dichloro-5-methylphenylthio)methyl)-1*H*-pyrazol-3(2*H*)-one (**1**) and 5-((4-chloro-2,5-dimethylphenylthio)methyl)-1*H*-pyrazol-3(2*H*)-one (**2**) (Figure 17), as chemical hits by means of HTS assay expressing mutant SOD1^G93A^. Then, 5-((3,5-dichlorophenylthio)methyl)-1*H*-pyrazol-3(2*H*)-one (**19**) (Figure 17), which was generated based on the structural optimization of ASP scaffold, was found to be more potent, with an EC_50_ value of 170 nM. According to pharmacokinetic assays, general parameters were amenable for compound **1**, except for its relatively rapid clearance and short microsomal half-life stability features. The optimization of ASP-based compounds stands out as novel therapeutic candidates for ALS treatment.

Nrf2/ARE (NF-E2-related factor 2/antioxidant response element) signalling program has been reported to be important in the regulation of oxidative damage, neuroinflammation and mitochondrial dysfunction. CDDO (2-cyano-3,12-dioxooleana-1,9-dien-28-oic acid) ethylamide (CDDO-EA) and CDDO trifluoroethylamide (CDDO-TFEA) (Figure 17) are two synthetic triterpenoid analogs derived from oleanolic acid. It was determined that these two oleanolic acid derivatives stimulated Nrf2/ARE in NSC-34 cell culture and in the SOD1^G93A^ ALS mouse model [207].

Cyclohexane-1,3-dione (CHD) derivatives were identified based on PC12-G93A-YFP HTS assay [208]. A more efficient compound with an EC_50_ value of 700 nM and more favourable pharmacokinetic profile, 5-(3,5-Bis(trifluoromethyl)phenyl)cyclohexane-1,3-dione (**26**) (Figure 17), was obtained via the structural modification of the CHD scaffold. However, compound **26** revealed no significant activity in the mutant SOD1^G93A^ mice model. These CHD analogs could be novel therapeutic candidates for further ALS studies.

A new GSK-3β inhibitor, 6-chloro-8-((3-(pyridin-4-yl)propyl)amino)-[1,2,4]triazolo[4,3-*a*]pyridin-3(2*H*)-one (JGK-263) (Figure 17) was developed, and its improved viability and neuroprotection was determined in normal and SOD1 wild/mutant NSC34 cell lines [209]. It was also observed that JGK-263 improved motor function and delayed the time until symptom onset and death in SOD1^G93A^ ALS mice model.

Tanaka et al. [210] documented that 2-[mesityl(methyl)amino]-*N*-[4-(pyridin-2-yl)-1*H*-imidazol-2-yl] acetamide trihydrochloride (WN1316) (Figure 17) selectively slowed oxidative stress-induced cell death and neuronal inflammation in the late-stage of ALS mice. WN1316, with high water solubility and BBB permeability, increased neuronal apoptosis inhibitory protein (NAIP) and Nrf2. Post-onset oral administration of low dose WN1316 improved motor function and extended the survival in SOD1^H46R^ and SOD1^G93A^ ALS mice models. The results of phase I clinical trial of WN1316 (UMIN000015054) have not been published yet.

Several ester and amide derivatives of chemical chaperones were synthesized by Getter et al. [211]. Among them, 3-((5-((4,6-dimethylpyridin-2-yl)methoxy)-5-oxopentanoyl)oxy)-*N*,*N*-dimethylpropan-1-amine oxide (**14**) (Figure 17) in vitro displayed both neuronal and astrocyte protection. Moreover, compound **14** improved the neurological functions and prolonged body weight loss in ALS mice.

## 3. Conclusions and Future Perspectives

ALS is a progressive and fatal motor neuron disorder. Moreover, it is a multifactorial syndrome rather than a single disease. There are several factors associated with the underlying pathogenesis of ALS, including oxidative stress (also associated with misfolding and aggregation of proteins), mitochondrial dysfunction, glutamate-induced excitotoxicity, apoptosis, neuroinflammation, axonal degeneration, skeletal muscle deterioration and viruses. Based on drug repurposing strategy, many drugs were tested in preclinical and clinical trials, but several of these trials have failed so far, owing to many reasons, such as (i) rareness of ALS causing limited participation of cohorts on trials, (ii) limited funds from pharma companies, (iii) disharmony between clinical and preclinical results, (iv) the complexity and heterogeneity features of the disease and (v) ambiguousness concerning the exact stage and region of the disease. Another main reason for the failure of clinical tests, as considered by many researchers, is as follows: When a clinical test begins, a high percentage of motor neurons have already dropped, and abnormal proteins are already accumulated in the motor neuron cell. (This is similar to Alzheimer’s disease, in which symptoms are not improved after removal of the amyloid). On the other hand, these obstacles profoundly encourage researchers to find an effective cure for ALS. Thus, many studies are underway that are devoted to this purpose, utilizing cutting-edge technologies. There is an urgent need to discover new therapeutics beyond drug repurposing methods to robustly combat ALS. Due to the variety of factors involved in the onset and progression of the disease, molecular hybridization, in which different agents can be exploited simultaneously on diverse targets, can open a new door in the treatment of ALS. In particular, riluzole and edaravone, approved drugs for the treatment of ALS, with their simple but active pharmacophores, attract great attention for the molecular hybridization approach. In addition, different kinase inhibitors such as Src/c-Abl inhibitors (dasatinib, imatinib, bosutinib) and masitinib and rapamycin were found effective on SOD1 mutant ALS mice models and performed well in clinical studies. Among them, Src/c-Abl inhibitors account for a large portion of this class, which is attributed to their wide range of effects on reducing apoptosis, boosting autophagy and suppressing neuro-inflammation, as previously discussed. Therefore, Src/c-Abl inhibitors may offer a prominent contribution to the generation of new hybrids as promising anti-ALS drug candidates.

## Figures and Tables

**Figure 1 ijms-23-02400-f001:**
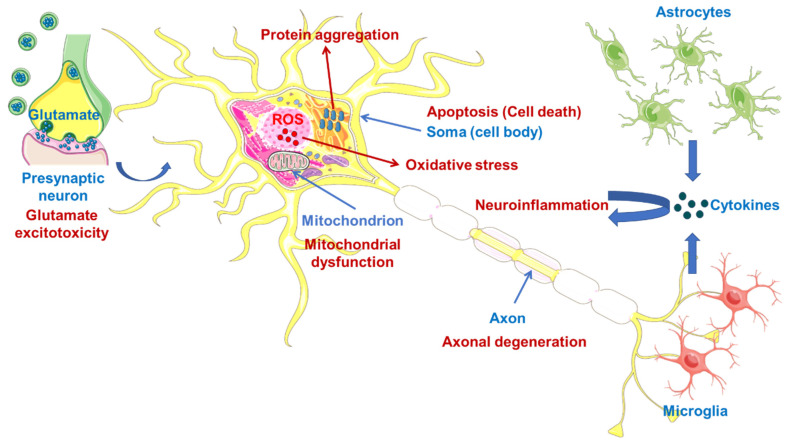
Some pathologic mechanisms in the central nervous system (CNS) related to the formation of ALS. Illustrations use elements from Servier Medical Art [28].

**Figure 2 ijms-23-02400-f002:**
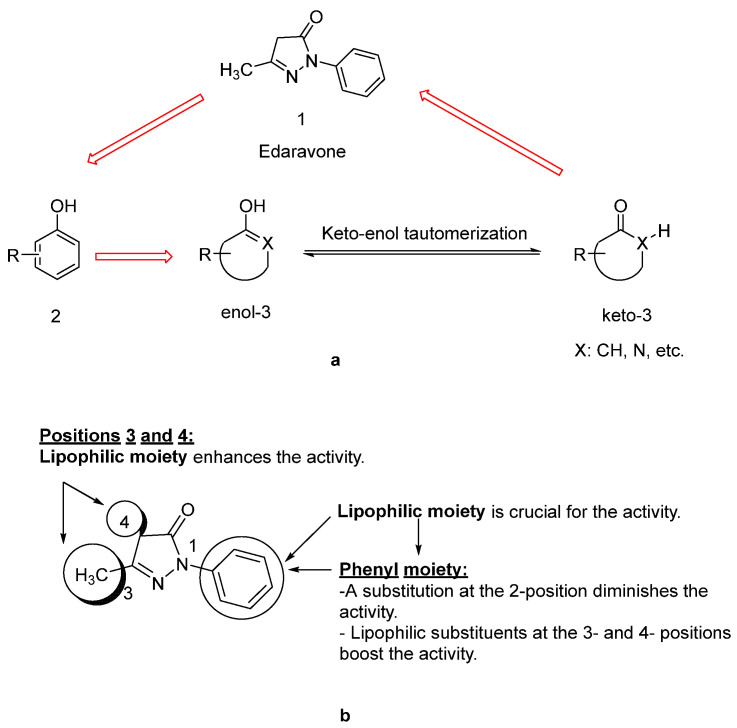
Design of a phenol-like compound (**a**), optimization of edaravone (**b**) [50,51].

**Figure 3 ijms-23-02400-f003:**
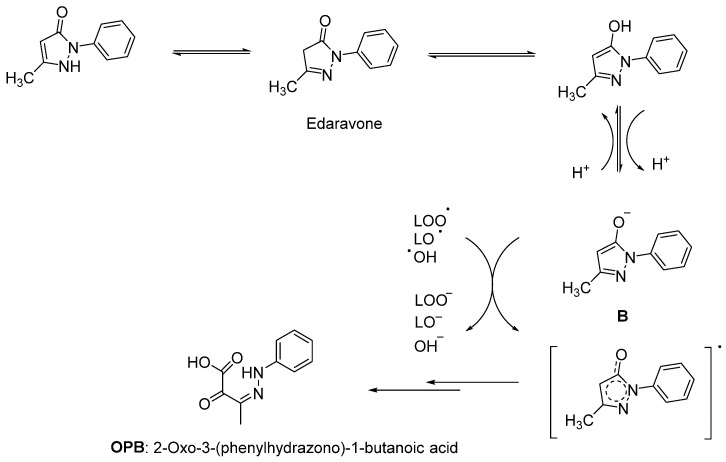
Reaction mechanism of edaravone with free radicals [52].

**Figure 4 ijms-23-02400-f004:**
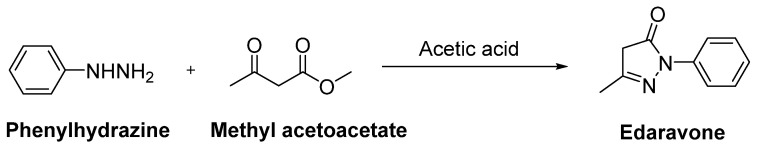
Synthetic route for edaravone.

**Figure 5 ijms-23-02400-f005:**
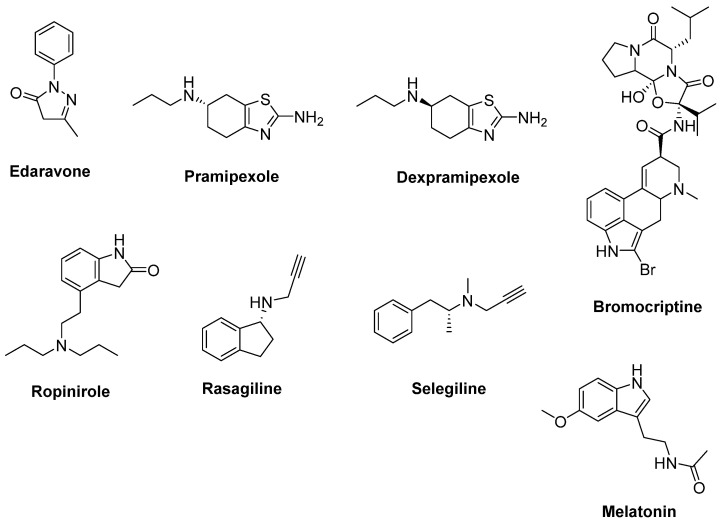
Therapeutics against oxidative stress.

**Figure 6 ijms-23-02400-f006:**
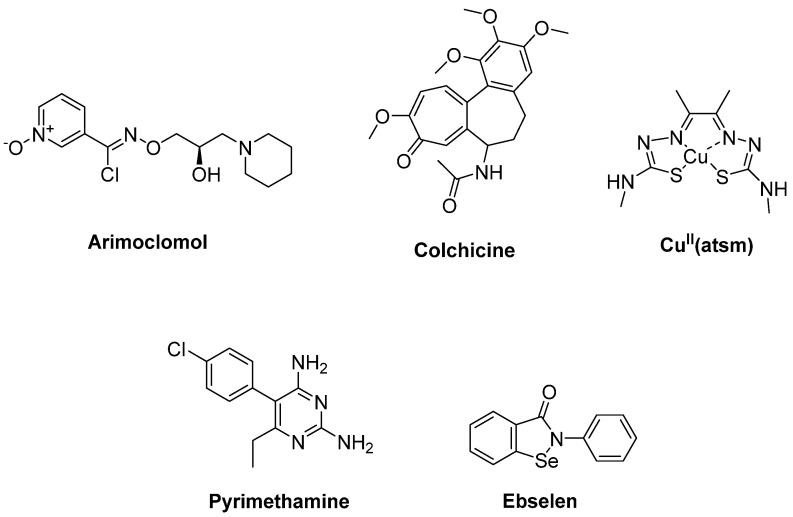
Therapeutics against oxidative stress via protein aggregation.

**Figure 7 ijms-23-02400-f007:**
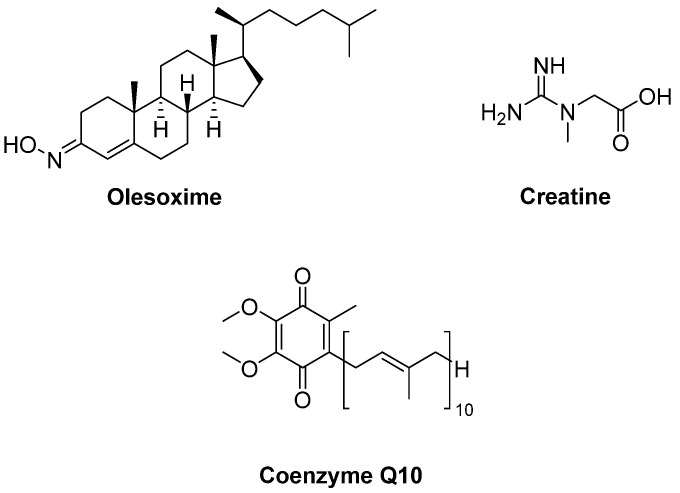
Therapeutics against mitochondrial dysfunction.

**Figure 8 ijms-23-02400-f008:**
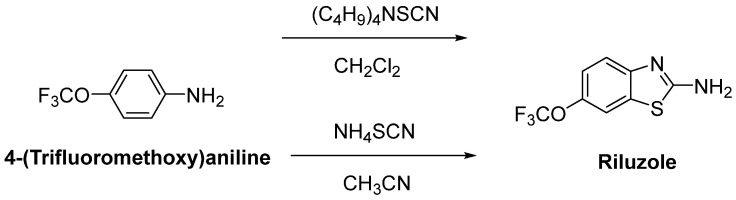
One-pot synthetic routes for riluzole.

**Figure 9 ijms-23-02400-f009:**
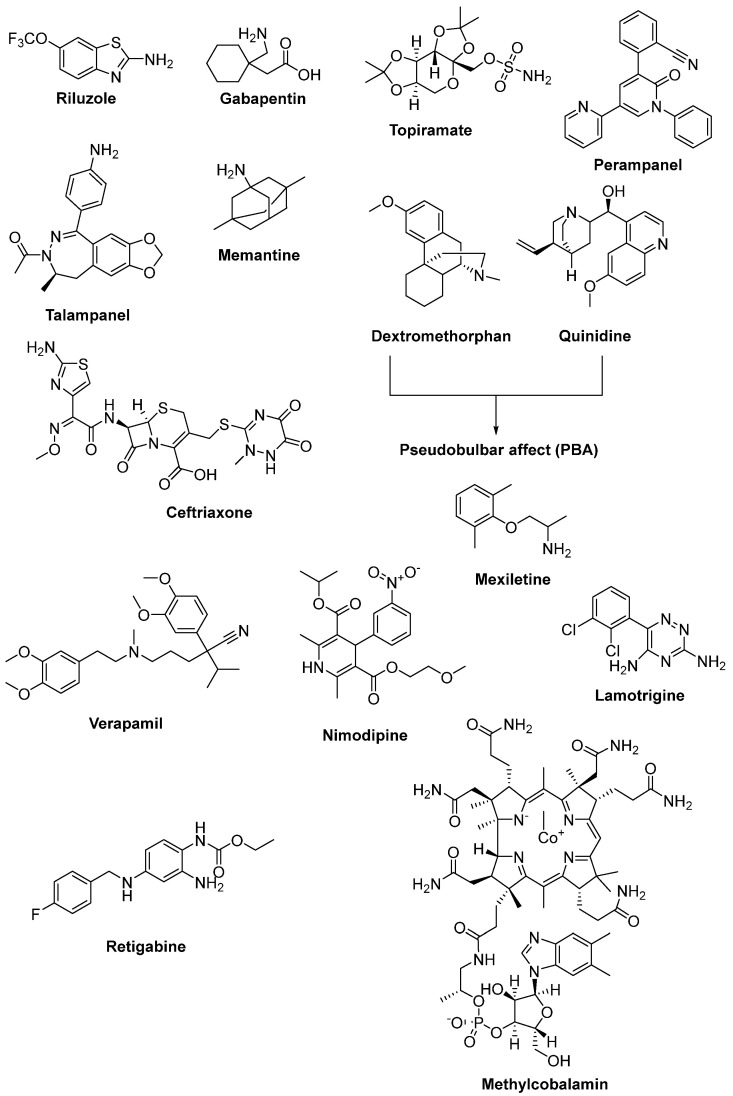
Therapeutics against glutamate-induced excitotoxicity.

**Figure 10 ijms-23-02400-f010:**
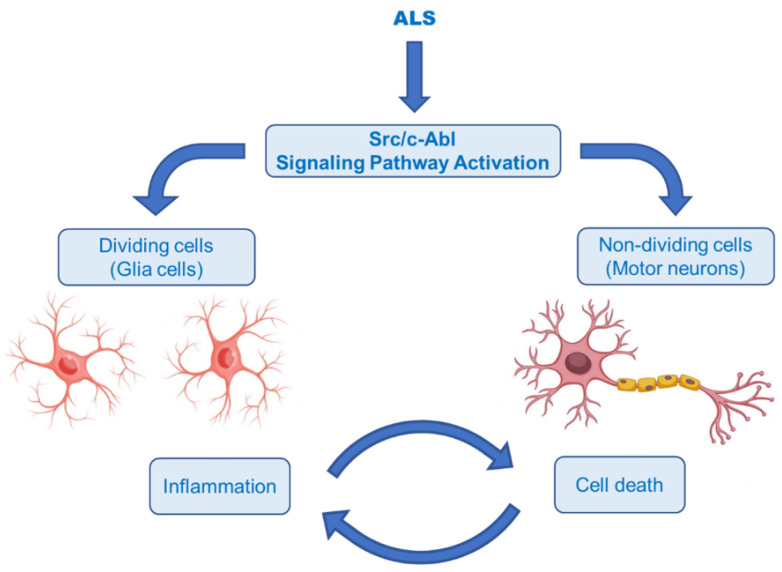
Hypothesis of the ALS pathomechanism in dividing and non-dividing cells [146]. Illustrations use elements from Servier Medical Art [28].

**Figure 11 ijms-23-02400-f011:**
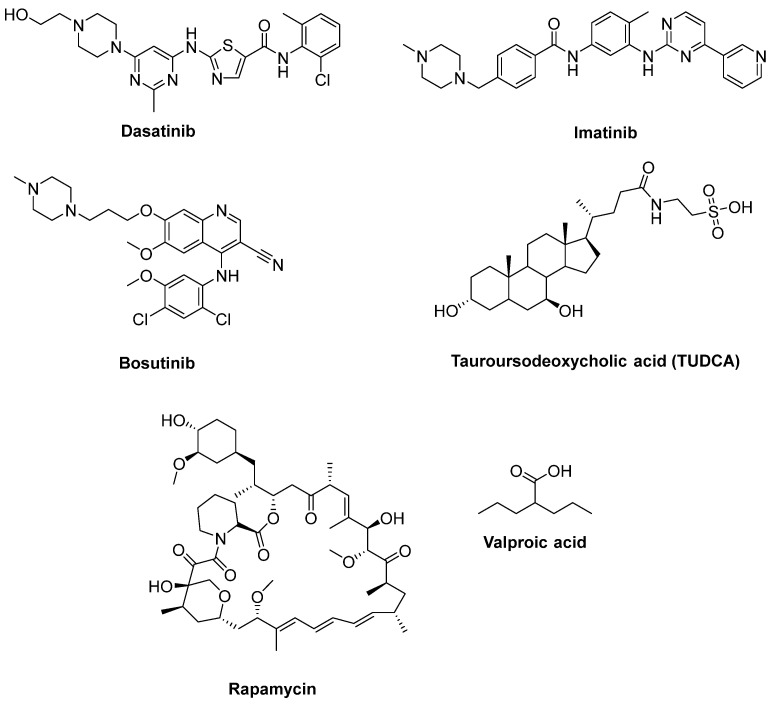
Therapeutics for reducing apoptosis and/or boosting autophagy.

**Figure 12 ijms-23-02400-f012:**
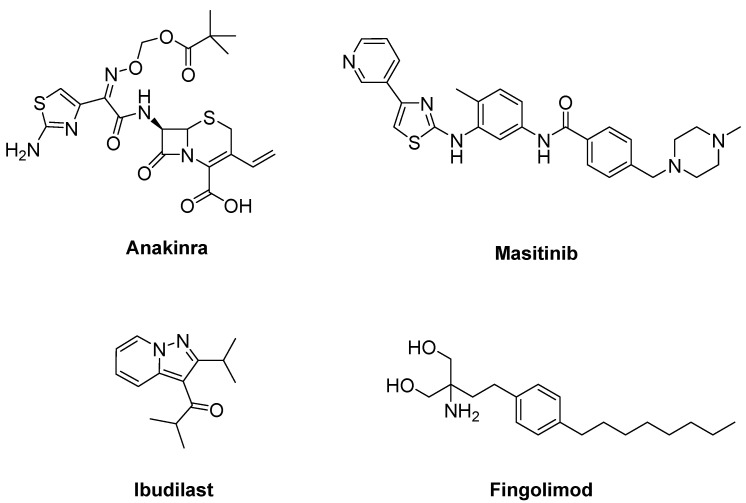
Therapeutics against for neuroinflammation.

**Figure 13 ijms-23-02400-f013:**
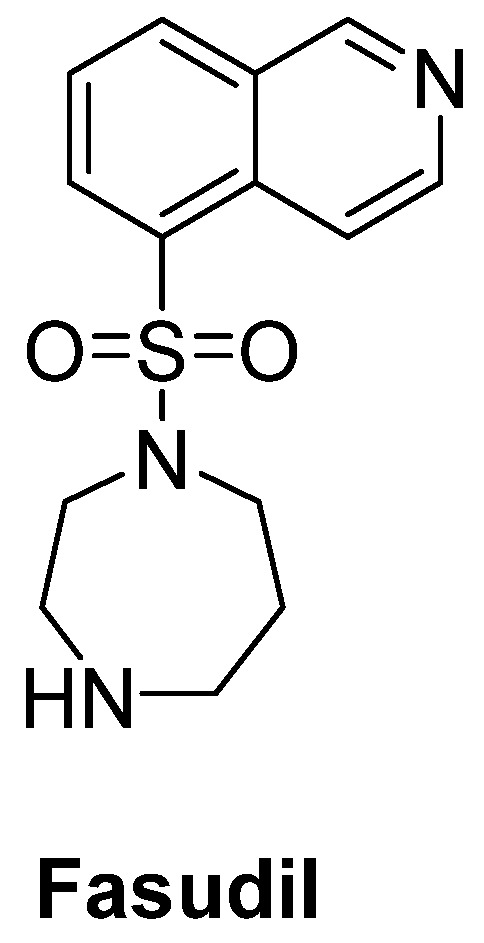
Fasudil, a promising ROCK inhibitor for alleviation of axonal degeneration.

**Figure 14 ijms-23-02400-f014:**
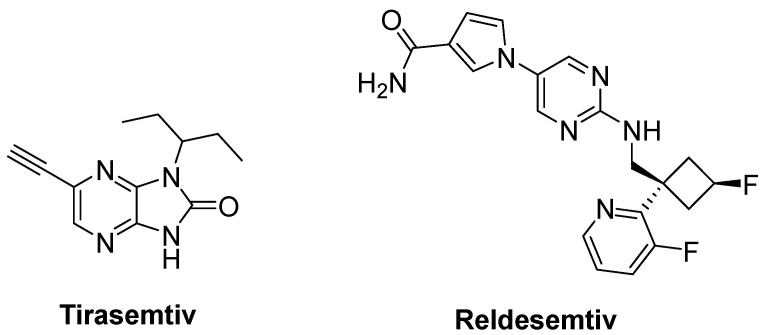
Therapeutics against skeletal muscle deterioration.

**Figure 15 ijms-23-02400-f015:**
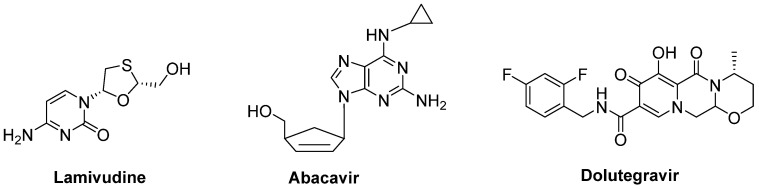
A potential combination of lamivudine, abacavir and dolutegravir to be effective in ALS.

**Figure 16 ijms-23-02400-f016:**
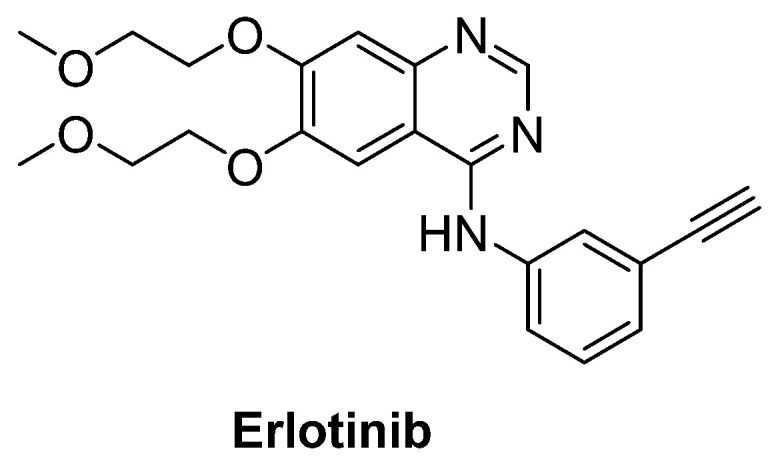
Erlotinib, an important EGFR inhibitor.

**Figure 17 ijms-23-02400-f017:**
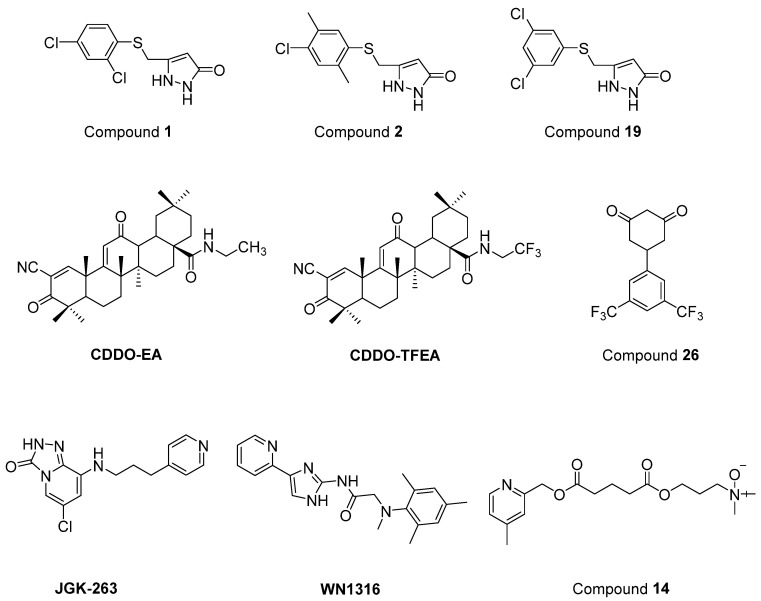
Newly synthesized and evaluated compounds as anti-ALS agents.

## Data Availability

Not applicable.

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
