# Peer review of "Comprehensive Research on Past and Future Therapeutic Strategies Devoted to Treatment of Amyotrophic Lateral Sclerosis"

_ijms, 2022, doi:10.3390/ijms23052400_

Round 1

Reviewer 1 Report

In this Review paper the authors present an in-depth study of currently available and developing treatments for Amyotrophic lateral sclerosis (ALS). The authors make a strong case that ALS is a complex desease and that this results in an alarmingly small number of drugs that passed clinical trials. They focused on therapeutic approaches targeting diverse mechanisms that are responsible for ALS pathogenesis.

The review is comprehensive, the writting style is very consize and therefore at times hard to read. There are 200+ cited references and although it was impossible to check all of them, they seem to be relevant. Nearly 50% of them were also published in the last 5 years.

The only point of concern for me is that the review is maybe too extensive, trying to sum up too many different treatment strategies. It is easy to get lost in countless studies and their results that are listed. Unfortunately, I don't think there is a fast and easy way to correct this.

To summarize, this Review presents a state of the art report on ALS treatment strategies and would - despite its considerable length - be of interest for wider audiences working on ALS and similar deseases. I therefore recommend its publication in the International Journal of Molecular Sciences.

Author Response

To the Editorial Office of IJMS and to the Editor.

Dear Dr. Battino, Dear Danna An,

We received your mail from February 3rd, 2022 with the comments and revision suggestions of the reviewers. We sincerely appreciate your decision and cordially thank you and all four reviewers for their constructive and valuable comments. We have addressed them to the best of our knowledge and conscious, revised the manuscript accordingly and have also further corrected it for the English language. We very much hope that the manuscript in this format will be better acknowledged by the journal audience and can now be considered for publication in IJMS.

Below please kindly find the reviewers comments and our point–by-point responses to them.

Reviewer1: In this Review paper the authors present an in-depth study of currently available and developing treatments for Amyotrophic lateral sclerosis (ALS). The authors make a strong case that ALS is a complex disease and that this results in an alarmingly small number of drugs that passed clinical trials. They focused on therapeutic approaches targeting diverse mechanisms that are responsible for ALS pathogenesis. The review is comprehensive, the writing style is very concise and therefore at times hard to read. There are 200+ cited references and although it was impossible to check all of them, they seem to be relevant. Nearly 50% of them were also published in the last 5 years. The only point of concern for me is that the review is maybe too extensive, trying to sum up too many different treatment strategies. It is easy to get lost in countless studies and their results that are listed. Unfortunately, I don't think there is a fast and easy way to correct this. To summarize, this Review presents a state of the art report on ALS treatment strategies and would - despite its considerable length - be of interest for wider audiences working on ALS and similar diseases. I therefore recommend its publication in the International Journal of Molecular Sciences.

 We sincerely appreciate the reviewer’s favorable remarks and valuable comments.

Reviewer 2 Report

The Article entitled “Comprehensive Research on Past and Future Therapeutic Strategies Devoted to Treatment of Amyotrophic Lateral Sclerosis”, submitted by Sever et al., summarizes the state of art in therapeutic approaches to treat ALS. Classifying drugs on basis of the cellular pathway they target, authors clearly describe each compound, reporting related benefits and limits in its use or potential use to treat ALS patients. Among all therapeutic strategies reported, they suggest Src/c-Abl inhibitors, targeting apoptosis pathway and boosting autophagy response, as the most promising anti-ALS drug candidates.

Few suggestions are reported in the attached file.

Author Response

Reviewer 2

The Article entitled “Comprehensive Research on Past and Future Therapeutic Strategies Devoted to Treatment of Amyotrophic Lateral Sclerosis”, submitted by Sever et al.,summarizes the state of art in therapeutic approaches to treat ALS. Classifying drugs on basis of the cellular pathway they target, authors clearly describe each compound, reporting related benefits and limits in its use or potential use to treat ALS patients. Among all therapeutic strategies reported, they suggest Src/cAbl inhibitors, targeting apoptosis pathway and boosting autophagy response, as the most promising anti-ALS drug candidates. In my opinion, this is a complete and well-done work.

We would like to thank the reviewer for his/her positive and constructive comments.We have modified the manuscript according to the comments, and detailed corrections are listed below.

However, I recommend an extensive editing of the paper and English revision.

  • The manuscript is thoroughly revised for language and grammatical mistakes.

I’m going to indicate some mistakes that need to be corrected: Lines 28, 43, 120, 169, 293: Please, correct typing errors.

  • Typing errors were corrected.

Lines 181-184: This part is confusing. Please correct appropriately.

  • The sentences in this paragraph were reformulated.

Line 194: What does “(B)” refer to?

  • The missing B was put into the Figure 3 (enolate form of edaravone )

Line 267: Please, correct “wel” into “well”.

  • Corrected

Lines 324 and 333: Please, indicate drugs’ name with the first capital letter, as reported in Figure 6. And so on…

  • The generic drug names are written with lower case letters in accordance with international standard rules. However, in all figures, the drug names are written with first capital letter, as indicated in each drug structure.

Reviewer 3 Report

In this manuscript the authors give a comprehensive review of pharmaceutical treatment strategies for ALS. They nicely show how potential drug candidates were identified in laboratory work and animal models of the disease and how attempts were made to assess these drug treatments in clinical studies. Their manuscript is organized along possible disease and therapeutic mechanisms in ALS.

The authors have done a very good job in following up how potential drug candidates were identified in basic research and then progressed into clinical trials. It is very helpful that the authors in their presentation also include the large number of clinical trials which yielded no positive outcome and failed. These negative data are often ignored and it is very helpful to mention them in this review article.

In their introduction and conclusion, the authors give a nice summary of the problems and possible chances in developing therapies for ALS. Taken together, this is a fine and helpful review article about the development of possible treatment options for ALS.

Author Response

Reviewer 3

In this manuscript the authors give a comprehensive review of pharmaceutical treatment strategies for ALS. They nicely show how potential drug candidates were identified in laboratory work and animal models of the disease and how attempts were made to assess these drug treatments in clinical studies. Their manuscript is organized along possible disease and therapeutic mechanisms in ALS. The authors have done a very good job in following up how potential drug candidates were identified in basic research and then progressed into clinical trials. It is very helpful that the authors in their presentation also include the large number of clinical trials which yielded no positive outcome and failed. These negative data are often ignored and it is very helpful to mention them in this review article. In their introduction and conclusion, the authors give a nice summary of the problems and possible chances in developing therapies for ALS. Taken together, this is a fine and helpful review article about the development of possible treatment options for ALS.

We thank the reviewer for his/her positive and constructive comments.

Reviewer 4 Report

In this review Sever and co-authors discuss old and new therapeutic strategies for the treatment of  Amyotrophic Lateral Sclerosis (ALS), particularly focusing on those ongoing or yet tested in ALS patients at different phases of clinical trials. The authors have structured the manuscript by describing the different strategies based on the different ALS pathogenic mechanisms targeted, which makes this Review very informative and clear.

Because of the great number of molecules discussed, to allow a better comprehension of the literature presented, the authors should add a Table summarizing the name, the mechanisms, and the state of the art of each therapeutic with proper references.

Minor points:

-In ALS, the classification M1/M2 microglia is not fully shared by the researchers. Please rephrase this concept on the basis of recent literature.

-English editing and correction of typos are needed.

Author Response

Reviewer 4

In this review Sever and co-authors discuss old and new therapeutic strategies for the treatment of Amyotrophic Lateral Sclerosis (ALS), particularly focusing on those ongoing or yet tested in ALS patients at different phases of clinical trials. The authors have structured the manuscript by describing the different strategies based on the different ALS pathogenic mechanisms targeted, which makes this Review very informative and clear. Because of the great number of molecules discussed, to allow a better comprehension of the literature presented, the authors should add a Table summarizing the name, the mechanisms, and the state of the art of each therapeutic with proper references.

Thank you for your valuable comments and suggestions on the structure of our manuscript. Point by point corrections are listed below

  • !  Due to the length of the manuscript and the large number of the drugs and their structures discussed, adding a large table, which would compile all drugs in the manuscript would exceed the frame of and will also adversely affect the fluency of the manuscript. Such a table can, however, be included into the Supplements, if you and the reviewer insists on it.

  • !  The classification M1/M2 microglia is rephrased in detail on the basis of recent literature.

    -English editing and correction of typos are needed.

    ! The manuscript is thoroughly revised for language and grammatical mistakes.

Minor points:

-In ALS, the classification M1/M2 microglia is not fully shared by the researchers. Please rephrase this concept on the basis of recent literature.
